# Monitoring audience engagement using electrodermal activity during an inaugural lecture

**Ivo V. Stuldreher** [1]*, **Anne-Marie Brouwer**[1,2]

**1** Human Performance, Netherlands Organization for Applied Scientific Research (TNO), Soesterberg, The Netherlands, **2** Artificial Intelligence, Donders Centre for Cognition, Radboud University, Nijmegen, The Netherlands

* ivo.stuldreher@tno.nl

## Abstract

Is an audience captured by a speech or lecture? At what times especially? Do different groups in an audience experience the same speech in different ways? Insight into attentional engagement of individuals can be valuable but difficult to quantify using self-report. Physiological synchrony, the degree to which physiological measurements such as electrodermal activity (EDA) of multiple people uniformly change, has been shown to covary with attentional engagement in lab settings. In this study, we moved out of the lab and monitored EDA of 30 individuals attending a real-life inaugural lecture. These individuals were labeled as belonging to either the *personal* or *professional* group, based on their relation with the speaker. We expected these groups to differ in their attentional engagement. We computed physiological synchrony between the participants and investigated how well this metric distinguished between the professional and personal groups, how well it marked predefined engaging events in the lecture, and its relation with levels of engagement as self-reported afterwards. Where possible, we compared physiological synchrony results to results based on individuals' EDA. We found that physiological synchrony in EDA can distinguish between the two groups. Individuals' EDA can also distinguish between the groups, if the occurrence and timing is known of an event that is expected to elicit different levels of engagement for the two groups. We further found that both synchrony and individuals' EDA measures mark predefined engaging events with above-chance accuracies. Neither was reliably related to self-reported levels of attentional engagement, highlighting the complementary value of EDA. Our work shows the sensitivity of EDA measures in real-life conditions, where low-level sensory effects, movement and speech cannot be the explanatory factor. Ultimate applications may be in educational and entertainment domains, exploring potential differences in attentional engagement patterns between experts and novices, or different target groups in entertainment.

**Data availability statement:** All data and scripts that allow reproduction of the results are available from the Open Science Framework through https://osf.io/q8c9b/.

**Funding:** Both authors of this paper were funded for the reported research by a TNO internal grant (iDeeSS 2023) and by the Dutch Ministry of Defense under grant number V2415. The funders did not play any role in the study design, data collection and analysis, decision to publish, or preparation of the manuscript.

**Competing interests:** The authors have declared that no competing interests exist.

## 1. Introduction

Presenting an audience with the same movie or a lecture does not mean that all audience members will experience the movie or lecture in the same way. Intrinsic interest and previous knowledge about the topic, or an instruction regarding what aspect is particularly important to attend to, will all determine individuals' engagement over time, and what exactly will be learned and memorized. We here investigate whether electrodermal activity (EDA) as recorded using wearables from real-life audience members, and especially the extent to which audience members show the same pattern, is informative of attentional engagement. With the term 'attentional engagement' we clarify that we do not only refer to focused attention, but also to active involvement in processing that information [1]. Quantifying and evaluating attentional engagement could support the development of effective educational material and entertainment. Real-time insight into attentional engagement may help to intervene when decreased engagement is detected, for instance by scheduling a break in a team briefing, or changing teaching style in an educational program. Quantifying differences in attentional engagement when attending a briefing between novices and experts could help evaluate level of expertise, and differences in attentional engagement between young and old audience members watching a movie could indicate that the experience is different for these two target groups.

Physiological measurements, such as the electroencephalogram (EEG), EDA or heart rate, may inform us on top-down determined attentional engagement – i.e., attentional engagement as determined by the cognitive or emotional state of the individual, aside from the basic sensory stimulation. For instance, individuals who attend to auditory stimuli show larger EEG event-related potentials in response to these sounds than individuals who are presented with these stimuli but are not attending [2]. Not only EEG, but also responses in peripheral physiological measures such as heart rate, EDA and pupil size may inform us on the attentional engagement of monitored individuals. These peripheral measures reflect arousal, the level of bodily activation. Attention and arousal are dynamically and intrinsically coupled [3], such that peripheral measures reflecting arousal can thereby also be informative of the attentional engagement of monitored individuals.

When aiming to use such event-related metrics in real-life situations, a challenge is that the type and timing of relevant events, required as an anchor for the event-related responses, are not always known. A promising method of monitoring attentional engagement without requiring information on events, is to assess physiological synchrony based on inter-subject correlations. Individuals that attend to the same narrative stimuli show synchronized neurophysiological responses, such as EEG, EDA or heart rate, that can be captured with inter-subject correlations [4,5]. Higher levels of inter-subject correlations are generally related to higher degrees of attentional engagement. For instance, individuals with higher inter-subject correlations answer more questions about the presented stimuli correctly [6,7], inter-subject correlations are higher when actively attending a stimulus than when focusing attention inward on a mental arithmetic task during stimulus presentation [8,9] and inter-subject correlations are particularly increased during moments in time that are expected to be engaging to the audience [10].

While inter-subject correlations in EEG generally seem to be most sensitive [5], inter-subject correlations in peripheral measures were found to predict attentional engagement as well [5,8,11]. This is important for the step from the laboratory to everyday settings, where the use of EEG may be challenging.

The method of inter-subject correlations does not depend on knowledge of events to relate responses to and does not require labeled training data that are hard to obtain in out-of-lab settings. Inter-subject correlations may be used to identify and support individuals that have difficulty attending to certain presented information or it may be used to detect moments in time that are experienced as engaging or disengaging. Still, there are very few studies actually demonstrating monitoring of attentional engagement using inter-subject correlations in out-of-lab settings. Dikker and colleagues recorded brain potentials of students in a controlled classroom using wearable EEG devices and found that brain-to-brain synchrony could be a possible marker of students engagement and social dynamics [12]. Gashi and colleagues monitored EDA and heart rate of pairs of presenters and audience members during conference presentations and found that synchrony between presenter and audience member predicted agreement on self-reported level of engagement [13]. A number of out-of-lab audience studies examined individuals' EDA responses, e.g., in the context of video advertisements [14], a symposium [15] and movies of dance performance [16]. These studies suggest that averaged individuals' EDA can inform us on experienced arousal or engagement by relating them to self-report [14,16] or observing that values are higher at the beginning of a talk and higher during interactive sessions [15]. EDA responses in these studies may be largely due to basic stimulus properties (sounds, visuals) and body movement rather than to top-down determined engagement. Furthermore, it is unknown whether inter-personal synchrony may be a more sensitive metric than individuals' EDA.

To the best of our knowledge, no study showed that inter-personal synchrony in peripheral measures can distinguish between individuals who are exposed to the same real-life event, but who are *a priori* expected to be differentially engaged.

Therefore, in the current work we monitor EDA of 30 audience members during an inaugural lecture, where half of the participants have a personal relationship with the speaker, and half a professional relationship. The type of relationship that audience members have with the speaker was chosen as a way to operationalize differences between individuals in attentional engagement over time, caused by a naturally occurring difference in individuals' background. Distinguishing between individuals with different relationships to the speaker serves as a proof-of-concept to test whether synchrony in EDA can distinguish between groups of individuals that are expected to differ in the attentional engagement. The lecture contains a-priori defined engaging events, part of which we expect to be more engaging for the group of individuals with a personal relationship with the speaker than the group of individuals with a professional relationship with the speaker. Our general aim is to identify whether interpersonal synchrony in EDA can be a marker of attentional engagement in a real-life setting, here an inaugural lecture. Besides synchrony-based metrics of EDA we also examine individual-based metrics of EDA where possible and compare their ability to provide information on attentional engagement.

We answer the following research questions:

- Can synchrony-based and individual-based EDA metrics distinguish between differentially engaged lecture audience members (personal versus professional relationship with the speaker)?

- How do synchrony-based and – individual-based EDA relate to a-priori defined engaging events in the lecture? Can these metrics be used to detect engaging events?

- Do synchrony-based and individual-based EDA relate to post-hoc, self-reported levels of audience engagement?

## 2. Materials and methods

### 2.1. Participants

30 participants (12 females), three younger than 30 years, 17 with an age between 30 and 50 years, and 11 older than 50 years, participated. Half of the recruited participants had a personal relationship with the speaker. The other half had a

professional relationship with the speaker. All participants provided written informed consent before taking part. The experiment was approved by Radboud University's ethical committee (ECSW-2023-078, reference 23N.007560) and conducted on the 5ᵗʰ of October 2023.

## 2.2. Materials

EDA was recorded at 32 Hz using EdaMove 4 wearables (Movisens GmbH, Karlsruhe, Germany). A picture of this wearable is shown in the top right window of Fig 1. The data were recorded from the palmar surface of the non-dominant hand using two solid gelled Ag/AgCl electrodes (MTG IMIELLA electrode, MTG Medizintechnik, Lugau, Germany, W55 SG, textured fleece electrodes, 55 mm diameter) and saved locally on the device. Heart rate was recorded at 1 Hz using Wahoo Tickr heart rate monitors (Wahoo Fitness LLC). Heart rate data were streamed over Bluetooth Low Energy to laptops that were placed in the room, where data were saved with custom-made software. Unfortunately, the Bluetooth connection of the majority of heart rate sensors failed, possibly due to interference in the lecture room. Therefore, heart rate could not be analyzed.

## 2.3. Stimuli

The data were recorded during a fifty-minute inaugural lecture. The full lecture can be found online at: https://www.youtube.com/watch?v=dBRgakLTmm4. In this lecture, the speaker introduced the research that she will focus on in her professorship. The lecture is meant for a broad audience, as it is attended both by colleagues with a similar academic background and by family and friends that do not necessarily have an academic background. The lecture included a-priori

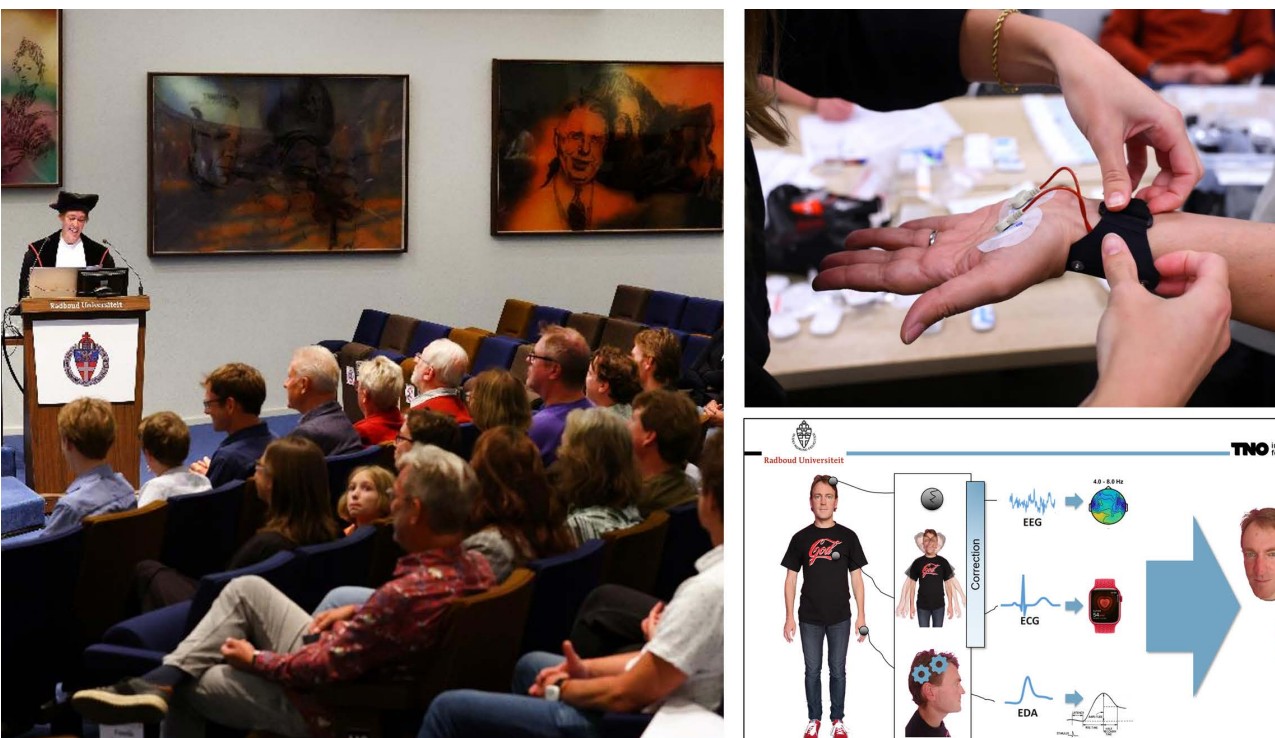

**Fig 1. Impression of the experiment.** Left: the setting in which the inaugural lecture took place. Top right: A Movisens EdaMove 4 sensor being applied to a participant. Bottom right: One of the slides of the inaugural lecture in which pictures of the speaker's brother were integrated. The identifiable individuals have given written informed consent (as outlined in PLOS consent form) to publish this figure.

defined events that could spark attentional engagement in either personally related audience, professionally related audience, or both. These events were the following:

*Walk-in.* At 02:06 minutes into the lecture (where the time corresponds to the online lecture as referred to above), the cortege including the speaker entered the room, during which the audience was asked to stand-up.

*Welcome.* At 03:12 the chair of the cortege welcomed the audience.

*Images of speaker's brother.* At multiple points during the lecture, namely at 7:44, 8:11, 9:05, 22:09, 22:34 and 40:11 images of the speaker's brother were shown in the slide deck, for instance as part of a graphic explaining how machine learning is used, or illustrating the concept of different mental states. An example of such a slide is shown in the bottom right window of Fig 1. The identifiable individuals have given written informed consent (as outlined in PLOS consent form) to publish this figure. Audience with a personal relation to the speaker would recognize the brother, while audience with a professional relation would not, resulting in a stronger expected attentional engagement in the former group.

*Instruction to stand.* 9:16 minutes into the lecture the speaker instructed the audience to stand up. This instruction was given in the context of demonstrating that besides mental activity also physical activity can strongly influence physiological measurements. It will also give us an impression of the effect of movement (added to mental attentional engagement associated with this task) on our measures of interest.

*Workload.* 10:34 minutes into the lecture the speaker instructed the audience to count the number of times she would use the letter *k* from now on until she instructed the audience to stop counting. 13:27 minutes into the lecture the speaker instructed that participants could stop counting and presented the number of *k*'s she had spoken.

*Sing-a-song stress-test.* 14:45 minutes into the lecture, the speaker showed a shortened version of the sing-a-song stress test [17]. In this paradigm, participants are shown a number of neutral sentences that are each followed by a 60 second countdown. Instead of a neutral sentence, the last sentence is an instruction to sing a song out loud once the 60 second countdown reaches zero. In the current experiment, participants were shown 2 neutral instructions (' think of how you travelled to this place' and 'think of what you normally have for breakfast') before the stress inducing instruction to sing a song once the timer reaches zero, where each of the three sentences were followed by a 10 second countdown.

*Personal video.* At 28:18 into the lecture, a picture of the speaker's son was shown on the slide. 30:07 minutes into the lecture, a video of the speaker's son, directed to the speaker's husband, was shown. This video was a demonstration of the loving videos paradigm, designed to elicit strong positive emotions in a laboratory setting [18]. The audience with a personal relationship would recognize the person as the speaker's son, while most of the audience with a professional relationship would not.

*Showing attentional engagement ranking.* During the lecture, the speaker had discussed that through measures of physiological synchrony one can gain insight in the attentional engagement of monitored individuals. 38:24 minutes into the lecture the speaker told the audience that a ranking regarding how well participants had attended to the lecture would be displayed on the presentation screen. After a minute the speaker revealed that such results were not available and would not be made public relating this to privacy and ethical issues around mental state monitoring.

*Acknowledgments.* Towards the end of the lecture, at 47:24, the speaker presented her acknowledgements.

## 2.4. Questionnaires

After the experiment participants answered a short questionnaire with questions regarding demographics, their understanding of the lecture and their attentional engagement during specific parts of the lecture.

A large part of the questionnaire was designated to capturing participants' self-reported level of engagement during specific lecture parts. We asked participants to rate their engagement on a 10-point Likert scale, ranging from 1 (very low) to 10 (very high). Participants did so for the following events: cortege entering the room and taking their places, welcome by the chair, pictures of speaker's brother, standing up, counting k's, sing-a-song stress test, personal video, engagement ranking, acknowledgments and lecture in general. In addition, they were asked to rate their emotional and cognitive engagement during the whole lecture.

## 2.5. Procedure

Participants were welcomed at the Radboud University and were guided to a separate room designated for the experiment preparation. Here, they first read the participant information document and signed an informed consent. Then, participants were assigned a participant number and were helped with attaching the sensors for monitoring EDA and heart rate. Participants were given instructions to sit in the middle seating area of the lecture hall where the inaugural lecture would take place. The lecture started by the speaker and a selection of attending professors walking in. A picture taken during the lecture is presented in the left window of Fig 1. After 50 minutes, the lecture ended and the speaker and attending professors left the room. Participants went back to the designated experiment room. Here, they first answered a short questionnaire, with questions regarding their demographics, their understanding of the interventions and their attentional engagement during parts of the lecture that were a-priori considered as potentially engaging. Then, participants were assisted with removing the wearable sensors.

## 2.6. Analysis

The data and scripts that allow reproduction of the results are available online at https://osf.io/q8c9b/. Data were analyzed using MATLAB 2023b (Mathworks, Natick, MA, USA).

**2.6.1. Pre-processing.** Artifacts in EDA were removed following earlier reported procedures [19,20]. First, parts of the data where the signal was below 0.3 µS were interpreted as suffering from a loose connection and marked as artifactual. This threshold was set lower than the 1 µS used in previous work, as visual inspection showed that a threshold of 1 µS would remove sound parts of data. Second, parts of the data surrounding data marked as artifactual or in between two segments of data marked as artifactual were also marked as artifactual. The marked data were replaced by NaN values. If more than 10 percent of the EDA of a participant were marked as artifactual, this participant was removed from further analyses. This resulted in the removal of data from four participants. From one other participant 0.8 percent of data were removed. No data were removed from the remaining 25 participants. In total, data of 26 participants (14 in the personal group and 12 in the professional group), are used in further analyses. The EDA data of these participants were smoothed using a third order Savitzky-Golay filter to remove quantization noise.

EDA was then processed to obtain the phasic component. To do so, we used continuous decomposition analysis as implemented in the Ledalab toolbox for MATLAB [21]. Note that performing continuous decomposition analysis on the EDA signal without removing artifacts first strongly impacts the result of the analysis of the parts surrounding artifactual data [19]. We therefore performed the continuous decomposition analysis separately on all segments in between parts of the data marked as NaN.

**2.6.2. Inter-subject correlations.** We then computed inter-subject correlations in the phasic component of EDA, following our previous work [5,8]. Summarizing, for each participant's phasic EDA, correlations were computed with the phasic EDA of all other participants in a 15 second moving window with a step size of 1 second. Averaging over the correlations with the phasic EDA of all other participants results in the participant-to-group inter-subject correlations. From now on, we refer to this metric as ISC-EDA.

**2.6.3. Significance of inter-subject correlations.** To test the significance of inter-subject correlations over chance level, we used the circular shuffle statistic, following [8,10,11]. Each participant's EDA was circular shifted by a random amount within the signal length. The ISC-EDA was then computed with this circular shuffled data. This procedure was repeated 1000 times for each participant to estimate the chance distribution of ISC-EDA. The p-value then is the fraction of circular shuffles with ISC-EDA values higher than the original unshuffled ISC-EDA.

**2.6.4. Inter-subject correlations in personal and professional relations.** We tested whether ISC-EDA differed between participants with a personal and professional relationship with the speaker. For each participant, we computed ISC-EDA with all other participants with a personal relationship and all other participants with a professional relationship to the speaker. Using paired-sample t-test we then investigated whether ISC-EDA differed when computed with other participants in the same or different group.

**2.6.5. Responses to engaging events in ISC-EDA and phasic EDA in personal and professional relations.** When information about the timing of events is available, an alternative approach to distinguish between groups is to analyze the response to an a-priori event that is expected to attentionally engage the groups differently. We here examined responses to the first time a picture of the speaker's brother was shown on the slide, as we hypothesized a larger response in participants with a personal relation to the speaker than for participants with a professional relationship to the speaker. For each participant, we extracted the amplitude of the response as the maximum value within 20 seconds after the event onset. We tested whether the amplitude was larger for participants with a personal compared to professional relationship to the speaker using an independent-sample t-test. Furthermore, we determined a value for classification accuracy as follows. A best guess of which participants belong to the personal group on the basis on responses, would be done by picking the the $N_{personal}$ participants with the highest response amplitudes to the picture, where $N_{personal}$ corresponds to the number of participants with a personal relationship to the speaker (i.e., 14). The percentage of these 14 participants with the highest response amplitudes that indeed belong to the personal group is then the classification accuracy. The abovementioned approach was performed for ISC-EDA and phasic EDA.

**2.6.6. ISC-EDA and phasic EDA to detect engaging events.** We investigated to what extent ISC-EDA and phasic EDA can be used for the detection of predefined hypothesized engaging events. We followed the approach that we introduced in [7], which uses signal detection theory to identify to what extent ISC-EDA and phasic EDA mark events that we hypothesized to be engaging, as described in the section 'stimuli'.

We marked the ISC-EDA and phasic EDA traces with time intervals during which the engaging events took place. During these times, an event detection would thus be considered correct (True) and an event detection during other moments would be considered incorrect (False). The moments in time where the trace is higher than a threshold t are marked as an event (Positive) and the moments in time where the trace is lower than t are marked as a non-event (Negative). Rather than using a single value for t, we use a gradually changing threshold t, ranging from the minimum to the maximum observed value of the trace. For each iteration of t, we can then define the true positives (TP), false positives (FP), true negatives (TN) and false negatives (FN). Using this, the true-positive rate or sensitivity (TPR) and the false-positive rate (FPR) or specificity can be computed. Plotting TPR against FPR provides the receiver operating curve (ROC). Detection performance was assessed using the standard metric of the area under the ROC (AUC of ROC).

We investigated whether detection of these events was higher than expected based on chance level by comparing the area under the receiver operator curve (AUC of ROC) to 1000 values obtained after randomly shuffling event markers in time.

**2.6.7. ISC-EDA and phasic EDA correlations with self-reported engagement.** To explore possible associations between post-hoc self-reported attentional engagement on the one hand, and ISC-EDA or phasic EDA on the other hand, we determined correlations between individuals' self-reports and averaged ISC-EDA or phasic EDA averaged across the corresponding event intervals.

## 3. Results

### 3.1. Distinguishing between audience with a personal and a professional relationship

Before investigating potential differences in ISC-EDA between participants with a personal or professional relationship with the speaker, we checked whether ISC-EDA were higher than expected based on chance level. Fig 2 depicts the ISC-EDA of all 26 participants with valid data compared to a chance distribution obtained using a circular shuffle approach. It shows that all 26 participants show ISC-EDA that are significantly higher than chance level.

We then investigated potential differences in ISC-EDA between participants with either a personal or a professional relationship with the speaker. Fig 3 shows that participants that have a personal relationship with the speaker have significantly higher within-group than between-group ISC-EDA ($t$ (13) = 2.32, $p$ = .037). For 11 out of the 14 participants with a personal relationship the within-group ISC-EDA are higher than the between-group ISC-EDA. Participants that have a professional relationship with the speaker do not have significantly higher within-group than between-group ISC-EDA ($t$ (11) = −0.04, $p$ = .970). For 7 out of the

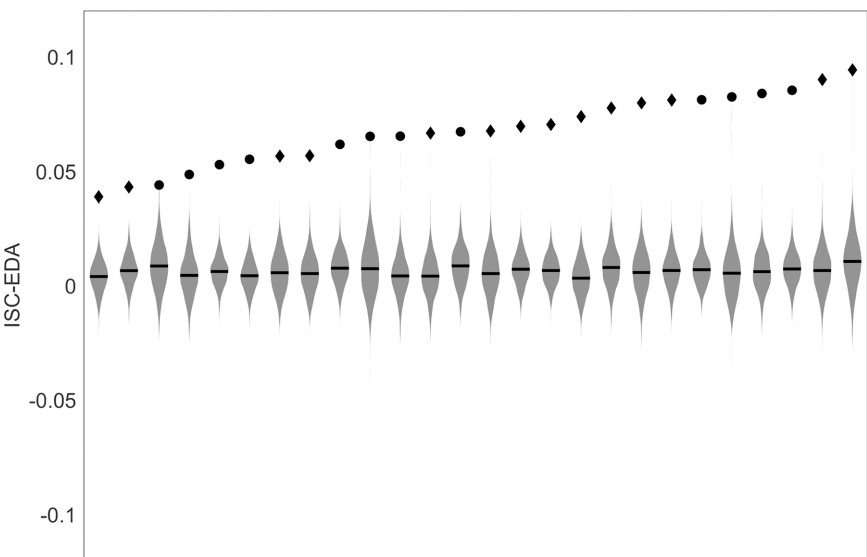

**Fig 2. ISC-EDA ordered form low to high compared to a chance distribution obtained after 1000 repetitions of circular shuffling.** Circles depict participants with a personal relationship to the speaker, diamonds depict participants with a professional relationship to the speaker. All ISC-EDA are significantly higher than chance.

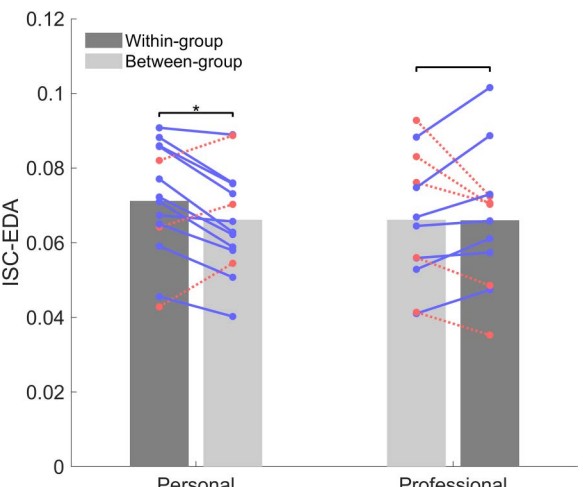

**Fig 3. ISC-EDA for participants with a personal or professional relationship with the speaker computing with participants belonging to the same (within-group) or different group (between-group).** Blue lines depict participants with higher within-group than between-group ISC-EDA, red dashed lines depict individuals with higher between-group than within-group ISC-EDA.

12 participants with a professional relationship the within-group ISC-EDA are higher than the between-group ISC-EDA. Classifying individuals based on the group with which they show highest physiological synchrony yields 69% classification accuracy.

An alternative group classification can be done by analyzing the response to an a-priori known event, in our case the first time the speaker's brother was shown on the slide. Fig 4 shows the response towards the first time the speaker's brother was shown on the slide for ISC-EDA and phasic EDA. For ISC-EDA, there is no significant response difference between participants with a personal or professional relationship to the speaker ($t(23) = 0.10$, $p = .922$). For phasic EDA,

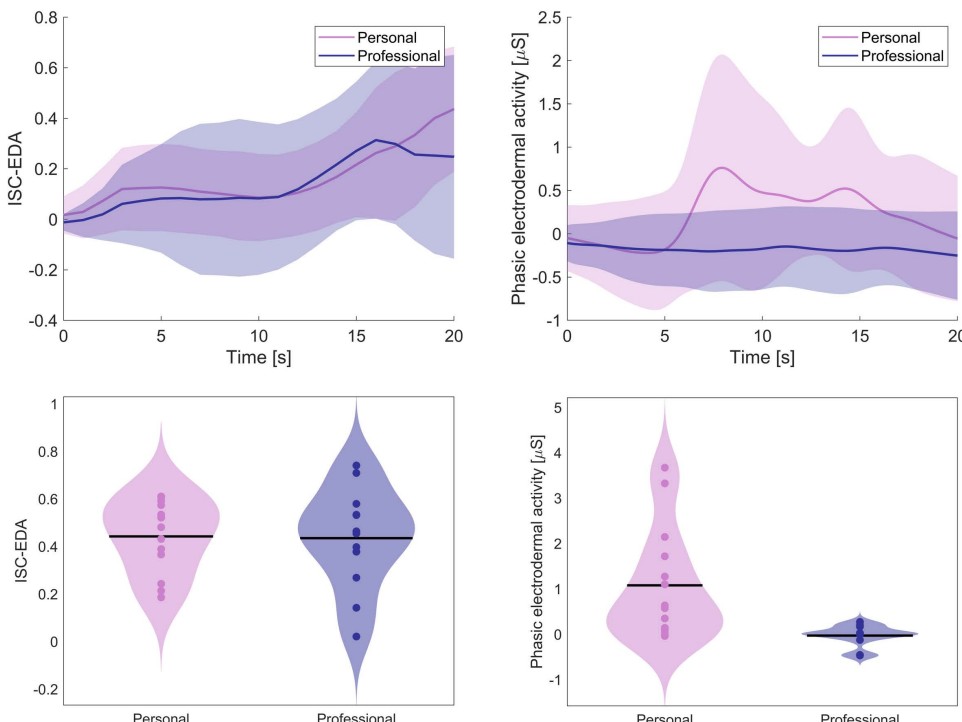

**Fig 4. ISC-EDA and phasic EDA for participants with a personal or professional relationship with the speaker in response to the first time the speaker's brother was shown on slide.** The top figures show the average response traces, the bottom figures show the individual response amplitudes.

participants with a personal relationship to the speaker have a significantly larger response to the event than participants with a professional relationship ($t(24) = 3.07$, $p = .005$). Classifying the top $N_{personal}$ participants as participants with a personal relationship to the speaker (i.e., the 14 highest data points in the lower panels in Fig 4) yields 71% of participants of which the relation is correctly identified for phasic EDA. For ISC-EDA, accuracy is at chance level (50%).

### 3.2. Detection of engaging events by ISC-EDA and phasic EDA

Fig 5 shows ISC-EDA and phasic EDA over time averaged across all participants. The figure suggests overlap between predefined engaging events and responses in both EDA measures. The instruction to stand up does not result in responses that are much higher than responses to engaging events that do not involve body movement. To investigate the sensitivity of ISC-EDA and phasic EDA to predefined events, we tested whether events could be detected based on these signals following our earlier work [7]. We used signal detection theory to quantify the accuracy with which events are detected and compared this to a chance distribution obtained by randomly shuffling event labels in time. For both ISC-EDA (AUC of ROC = .658) and phasic EDA (AUC of ROC = .703), the AUC of ROC scores were higher than AUC of ROC scores obtained on datasets where the event markers were shuffled in time (ISC-EDA: $ROC_{rand} = .498 \pm .066$, $t(999) = 2.30$, $p = .022$; phasic EDA: $ROC_{rand} = .485 \pm .069$, $t(999) = 3.14$, p = .002).

### 3.3. Correlations with self-reported engagement

We asked participants to rate their attentional engagement during the entire lecture and during the individual events in the lecture. We computed correlations between ISC-EDA or phasic EDA (averaged over the intervals indicated in Fig 4) and these self-reported engagement scores. Table 1 shows the results. We found no

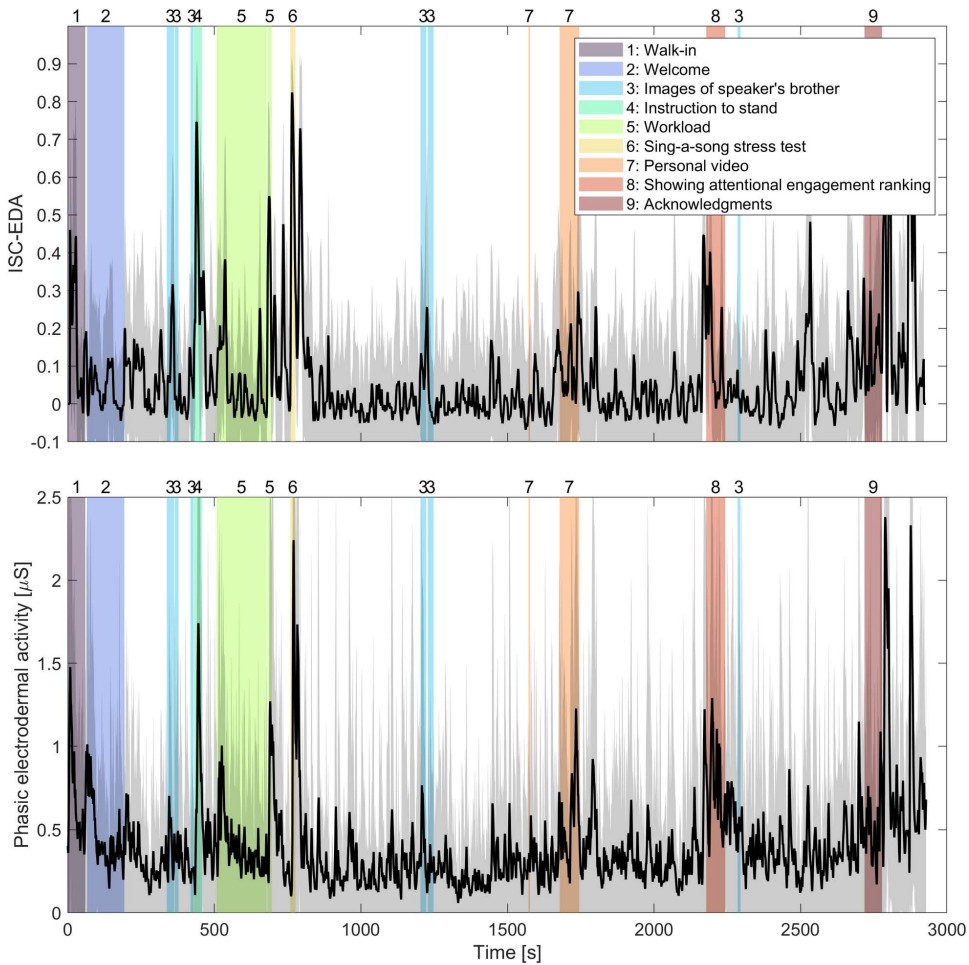

**Fig 5. ISC-EDA (A) and phasic EDA (B) averaged across participants over time.** Grey shading around the lines correspond to the standard deviation across participants. Vertical colored and numbered bars show the predefined events.

**Table 1. Correlations between self-reported ratings of attentional engagement and ISC-EDA or phasic EDA.**

|  | ISC-EDA | Phasic EDA |
|---|---|---|
| Overall engagement | r = −.11, p = .590 | r = −.14, p = .515 |
| Cortege entering | r = .07, p = .730 | r = −.19, p = .341 |
| Welcome | r = −.13, p = .524 | r = −.08, p = .709 |
| Image of speaker's brother | r = .16, p = 455 | r = .09, p = .669 |
| Instruction to stand | r = −.11, p = .613 | r = −.17, p = .401 |
| Workload | r = −.13, p = .526 | r = −.22, p = .275 |
| Sing-a-song stress test | r = −.13, p = .522 | r = −.17, p = .393 |
| Personal video | r = .20, p = .346 | **r = .39, p = .049** |
| Showing attentional engagement ranking | r = .08, p = .709 | r = −.37, p = .066 |
| Acknowledgments | r = .02, p = .930 | r = −.13, p = .529 |

significant correlation between ISC-EDA and the general reported attentional engagement ($r = -.11$, $p = .590$) or phasic EDA and the general reported attentional engagement ($r = -.14$, $p = .515$). When investigating correlations during ratings of specific events and the ISC-EDA and phasic EDA levels during those events, we found one significant correlation between phasic EDA and self-reported engagement for the personal video of the speaker's son ($r = .39$, $p = .049$).

## 4. Discussion

In the current work we investigated the relation between attentional engagement and EDA -both inter-subject correlations in phasic EDA (ISC-EDA) and individuals' phasic EDA – in participants attending a real-life inaugural lecture. Our results indicated that both ISC-EDA and phasic EDA can be informative of the attentional engagement of monitored individuals. Below, we discuss the research questions regarding distinguishing between participant categories, detecting events in time and relation to self-reported attentional engagement separately.

### 4.1. Distinguishing between participant categories

Our first research question concerned distinguishing between audience with a personal or professional relationship to the speaker, exemplifying classes of individuals that we expect to have different patterns of attentional engagement. If no information is available on the exact presence and timing of events that would distinguish between classes of individuals, ISC-EDA is the only marker that can be used to distinguish between groups. ISC-EDA allowed distinguishing between the two participant categories with 69% accuracy. Audience with a personal relationship to the speaker showed significantly larger inter-subject correlations with individuals that also had such a relationship instead of a professional relationship to the speaker. For audience with a professional relationship, we did not find such a difference. This may be caused by the broad diversity of participants in the professional relationship group. Some of the participants designated to this group had a relatively close personal connection with the speaker, while others had a more distant connection. This may thus have caused more diverse reactions to the stimuli. Participants designated to the professional group but with a relatively close personal connection with the speaker may have responded in a similar way to stimuli as participants with a personal connection.

When using information about the timing of an event that would likely elicit differential attentional engagement in the two participant groups, phasic EDA allowed for distinguishing between groups as well. Computing the phasic EDA amplitude following a moment in the lecture where the speaker's brother was shown allowed for distinguishing between audience with professional or personal relation to the speaker with 71% accuracy. Applying the same approach to ISC-EDA resulted in chance level accuracy. If available, using contextual information can support distinguishing between groups of individuals expected to differ in patterns of attentional engagement.

The effect of groups shows that EDA measures are sensitive enough in real-life environments to pick up effects of attentional engagement that cannot be explained by sensory factors or body movement.

### 4.2. Detecting events in time

The second research question of the current work concerned identifying predefined engaging events in time using ISC-EDA and phasic EDA. Using signal detection methods as previously done in [7] we found that both ISC-EDA and phasic EDA could detect predefined engaging events with accuracies above chance level. Again, the fact that we find positive results in this out-of-lab settings vouches for the robustness of the methodology and the signals' relation with attentional engagement. Though we a-priori expected the selected events to be engaging compared to the remainder of the lecture, the 'true' pattern of attentional engagement for each participant in this dynamic out-of-lab setting is unknown. Some 'engaging' events may not have been so engaging after all, while there could have been additional moments that

were experienced as engaging which were not marked as such in our analysis. We show that even under these dynamic settings, without exact knowledge of ground truth attentional engagement, predefined engaging events could be detected with accuracies higher than chance.

### 4.3. Relation to self-reported engagement scores

Our third and final research question concerned the relation between ISC-EDA and phasic EDA with self-reported levels of engagement during the lecture and during specific parts of this lecture. For both ISC-EDA and phasic EDA the relation with self-reported levels of engagement was limited, with no significant relations between ISC-EDA and self-reported levels of attention and only one significant relation between phasic EDA and self-reported levels of attention. On its own, this null result could have indicated that our physiological metrics do not reflect attentional engagement. However, considering our other positive results and the observation that previous work has consistently shown a relation between inter-subject correlations and stimulus retention as an objective measure of attentional engagement for EDA [22,23], HR [5,8,10,22], and other physiological metrics [11], we take this as an illustration of the importance of implicit physiological measures as metric of attentional engagement. Full information about attentional engagement cannot always simply be obtained using post-hoc questionnaires. Similarly, while Gashi et al. [13] find a positive relation between speaker-listener synchrony for certain EDA measures on the one hand, and *agreement* on self-reported engagement between speaker and listener on the other hand, they do not find an association between EDA synchrony and self-reported engagement by listeners. Gashi et al. attribute this to the size of the available dataset. In general, we also expect a positive relation between self-reported engagement and synchrony, but self-report may not be a very sensitive measure. Individuals may find it hard to rate their level of engagement, especially when such ratings are only obtained well after the to-be-rated moments in time. Note that obtaining ratings closer in time, i.e., during the lecture, would have interfered with their attentional engagement itself. Also note that the specific case under study (a festive, inaugural lecture by one of the researchers) may have led to response bias, participants being hesitant to report low engagement.

### 4.4. Conclusions

Overall, we show that EDA as recorded using a wearable in a real-life setting can be informative of the attentional engagement of individuals in a group. ISC-EDA and phasic EDA can both distinguish between individuals expected to differ in patterns of attentional engagement and they can both be used to identify engaging moments in time. A major advantage of ISC-EDA over phasic EDA is that contextual information about events is not necessary to identify individuals sharing attentional focus. For identifying engaging moments in time, the simpler measure of phasic EDA seems more suitable than ISC-EDA.

### Acknowledgments

We thank Juliette Bruin and Paola Perone for their help in the data collection. We also thank Movisens GmbH for lending us 13 EdaMove 4 sensors for this experiment.

### Author contributions

**Conceptualization:** Anne-Marie Brouwer.

**Data curation:** Ivo V. Stuldreher.

**Formal analysis:** Ivo V. Stuldreher.

**Funding acquisition:** Anne-Marie Brouwer.

**Investigation:** Ivo V. Stuldreher.

**Methodology:** Ivo V. Stuldreher, Anne-Marie Brouwer.

**Project administration:** Anne-Marie Brouwer.

**Resources:** Anne-Marie Brouwer.

**Software:** Ivo V. Stuldreher.

**Supervision:** Anne-Marie Brouwer.

**Validation:** Ivo V. Stuldreher.

**Visualization:** Ivo V. Stuldreher.

**Writing – original draft:** Ivo V. Stuldreher, Anne-Marie Brouwer.

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
