## [Decision Letter · Decision Letter 0]

Dear Dr. Stuldreher,

Thank you for submitting your manuscript to PLOS ONE. After careful consideration, we feel that it has merit but does not fully meet PLOS ONE’s publication criteria as it currently stands. Therefore, we invite you to submit a revised version of the manuscript that addresses the points raised during the review process.

**The reviewers highlighted the need for greater clarity in defining participant groups and the methodology, The novelty of the study should also be better claimed, distinguishimg between attentional engagement and interpersonal connection. Additionally, the manuscript should improve the discussion on the relationship between physiological synchrony and self-reported engagement. Please prepare a major revision accordingly, following the reviewers’ detailed comments.**

We look forward to receiving your revised manuscript.

Kind regards,

Alberto Greco

Academic Editor

PLOS ONE

**Journal Requirements:**

2. We note that Figure 1 includes an image of a participant in the study. 

Reviewers' comments:

Reviewer's Responses to Questions

**Comments to the Author**

1. Is the manuscript technically sound, and do the data support the conclusions?

Reviewer #1: Partly

Reviewer #2: Partly

2. Has the statistical analysis been performed appropriately and rigorously?

Reviewer #1: Yes

Reviewer #2: Yes

3. Have the authors made all data underlying the findings in their manuscript fully available?

Reviewer #1: Yes

Reviewer #2: Yes

4. Is the manuscript presented in an intelligible fashion and written in standard English?

Reviewer #1: Yes

Reviewer #2: Yes

**Reviewer #1:**  Understanding audience engagement using physiological signals and synchronization is an interesting topic. However, I found the manuscript challenging to follow.

Comments:

• In the abstract, “investigated how well it distinguished between these groups of participants” is unclear because the groups were not previously defined. It seems the grouping is based on personal or professional relationships with the speaker, but this is not explicitly stated. Additionally, some participants could have both types of relationships.

• The statements “We compare the outcomes to results” and “Unlike analyses using individual’s EDA, analyses with inter-subject correlations do not require contextual event information” are unclear and need clarification.

• It seems that the main finding is that EDA synchrony is not effective at detecting attentional engagement but is better at capturing a kind of “empathy” or relationship with the speaker. I suggest reflecting this more consistently throughout the paper, including the title. The results for engagement detection appear to be secondary in the analysis and are also not very positive.

• Have the authors considered analyzing the correlation of the tonic component of EDA?

• Section 2.6.2 suggests that the correlation of the SCRs (a more accurate term would be “phasic component”) was computed. However, subsequent sections about correlation analysis refer to participants’ EDA. Were raw EDA and phasic EDA correlations computed? Please clarify this.

• I question the practical use of a system designed to capture the type of relationship (personal or professional) an audience member has with the speaker. It is uncommon to give a lecture primarily to family and friends. Please elaborate on the potential applications of this system.

• I have not used EdaMove 4, but the statement about it being a “high quality” wearable seems unnecessary and unsupported by the data presented.

• ISC-EDA was never defined. Initially, I thought ISC referred to the correlation, but since it was computed using the phasic component, it was labeled SCR. However, these terms appear to differ. It is unclear what SCR represents in the text, tables, and figures, as it does not seem to correspond to the phasic component of EDA for an individual.

• I initially thought the correlation was computed between the speaker and the participants, but it seems to be inter-subject. Furthermore, it appears that the correlation was performed within-group (same group) and between-group (other group). However, these groupings were not defined in the text and are not intuitive. This aspect of the manuscript was particularly difficult to follow.

• The statement “when we refer to inter-subject correlations, we refer to this participant-to-group metric” is confusing because the terms are used later in the text as if they represent different concepts: “The inter-subject correlations and participant-to-group inter-subject correlations.” The manuscript, especially the methodology, needs significant improvement to provide greater clarity.

**Reviewer #2: ** This study investigates physiological synchrony in electrodermal activity (EDA) as a real-time marker of attentional engagement in a real-life lecture setting. Results show that both physiological synchrony and individual EDA can differentiate between participant groups and detect engaging events but do not reliably align with self-reported engagement.

The study is overall well-executed and the analyses are comprehensive and detailed. Nevertheless, I believe that the novelty and contribution to the field may be not sufficient for a publication (as detailed below). A more detailed framing highlighting the novelty of this study may be useful to reach publication standards.

INTRODUCTION:

- MAJOR: The Introduction could be further expanded and elaborated.

- MAJOR: The overall novelty of this study should be highlighted more, as (at the current state) it does not seem to bring a significant contribution to the field.

- MAJOR: A general aim should be presented before specific research questions.

- MINOR: Please clarify the role and meaning of peripheral measures, before explaining how they actually relate to attentional engagement.

METHODS:

- MINOR: Is there any particular reasons why the Bluetooth connection of the majority of heart rate sensors failed and not EDA sensors? This could be useful for future research as well.

- MAJOR: The event "pictures of Radboud colleagues" is used in self-report questionnaires but is not reported in events of interest. Could you specify the timing of this event? Was it used during the signal analysis? The analysis in 2.6.5 should be repeated on this event expecting opposite results compared to the brother's picture.

- MINOR: In Procedure: "attending professors leaving the room" -> "left"

- MAJOR: Since the EDA signal is often varying in time trajectory according to individual differences, what do the authors think about using lag-based measures, such as cross-correlation?

- MINOR: The acronym "ISC-EDA" is never explicitly defined in the main text.

- MINOR: Please define more details in the classification approach described t the end of paragraph 2.6.5.

- MINOR: 2.6.6; also this analysis should be clarified.

- MINOR: Analyses of self-report questionnaires are not specified in the Methods.

- MINOR: A public version of the dataset should be made available, at least for the Reviewers (there is a specific option in OSF). Until now, it was possible to access it only as contributor.

RESULTS:

- MINOR: "highest physiological synchrony with yields 69%" -> "which"

- MINOR: "Figure 2 shows that participants that have a personal relationship with the speaker have significantly..." is actually Figure 3.

DISCUSSION:

- MAJOR: Please discuss the absence of a relation between physiological synchrony and self-report measures in light of Gashi et al.'s [12] results.

- MAJOR: most results (except for the event detection) are about the SCR and not ISC-EDA. Showing that the phasic component of EDA is related to attentional engagement is not an overwhelming result, so the novelty of this study should be highlighted more clearly.

- MINOR: Fig. 4, low panels: adding a boxplot or violin plot to the points could be useful to visualize the results.

**Do you want your identity to be public for this peer review?** For information about this choice, including consent withdrawal, please see our Privacy Policy

Reviewer #1: No

Reviewer #2: No

---

## [Author Response · Author response to Decision Letter 1]

25 Mar 2025

Dear Dr. Greco,

We would like to thank you and the reviewers for your work and feedback on our mansucript. It helped us to improve the manuscript considerably. Below are the detailed point-by-point responses to the reviewers, but overall, the most important changes are:

- Clarification of the added value of the current work in the Introduction and Discussion.

- Clarification of the methods and terminology used.

- Clarifying the rationale of the participants groups – we explain that this classification was used to operationalize expected differences in attentional engagement (in this case, due to the stronger or weaker personal connection with the speaker).

- We discuss the relation between self-report and physiological synchrony more elaborately.

Furthermore, we obtained written permission from the people who can be recognized in the picture, to publish the picture.

On behalf of all authors,

Best regards,

Ivo Stuldreher

---

## [Decision Letter · Decision Letter 1]

Monitoring audience engagement using electrodermal activity during an inaugural lecture

PONE-D-24-58554R1

Dear Dr. Stuldreher,

We’re pleased to inform you that your manuscript has been judged scientifically suitable for publication and will be formally accepted for publication once it meets all outstanding technical requirements.

Kind regards,

Alberto Greco

Academic Editor

PLOS ONE

Additional Editor Comments (optional):

Reviewers' comments:

Reviewer's Responses to Questions

**Comments to the Author**

Reviewer #1: All comments have been addressed

Reviewer #2: All comments have been addressed

2. Is the manuscript technically sound, and do the data support the conclusions?

Reviewer #1: Yes

Reviewer #2: Yes

3. Has the statistical analysis been performed appropriately and rigorously?

Reviewer #1: (No Response)

Reviewer #2: Yes

4. Have the authors made all data underlying the findings in their manuscript fully available?

Reviewer #1: Yes

Reviewer #2: Yes

5. Is the manuscript presented in an intelligible fashion and written in standard English?

Reviewer #1: Yes

Reviewer #2: Yes

Reviewer #1: I want to thank the authors for properly addressing my comments and suggestions. The manuscript is a better product.

Reviewer #2: All my comments were adequately addressed. Many parts of the manuscript were modified accordingly, or choices were properly justified.

I would like to thank the authors for their thorough work on improving this manuscript.

Only a couple minor (grammatical) issues persist in section 2.6.5, and should be corrected before publication:

"personal group on the basis on responses" -> "basis OF responses"

"would be done by picking the the Npersonal participants" -> remove one THE

**Do you want your identity to be public for this peer review?** For information about this choice, including consent withdrawal, please see our Privacy Policy

Reviewer #1: No

Reviewer #2: **Yes: ** Francesco Bossi

---

## [Editor Report · Acceptance letter]

PONE-D-24-58554R1

PLOS ONE

Dear Dr. Stuldreher,

I'm pleased to inform you that your manuscript has been deemed suitable for publication in PLOS ONE. Congratulations! Your manuscript is now being handed over to our production team.

Kind regards,

on behalf of

Dr. Alberto Greco

Academic Editor

PLOS ONE